# Development of Data Transfer Ethics Framework (daTEF): A participatory approach to delivering evidence-based guidelines for healthcare data transfer

Gopiram Syangtan[1], Sauhardra Manandhar[1], Manjula Bhattarai[1], Binu Shah[1], Minu Singh[1], Binod Rayamajhee[1,2,3]*, Anurag Adhikari[1]

**1** Department of Infection and Immunology, Kathmandu Research Institute for Biological Sciences, Lalitpur, Bagmati, Nepal, **2** School of Optometry and Vision Science, Faculty of Medicine and Health, UNSW, Sydney, Australia, **3** Faculty of Medicine, Health and Human Sciences, Macquarie University, Sydney, Australia

* rayamajheebinod@gmail.com, binod.rayamajhee@mq.edu.au

## Abstract

### Background

In Nepal, insufficient healthcare infrastructure and limited funding contribute to unmet public healthcare needs and reduced quality of care. While foreign health researchers have stepped in to support local research initiatives, their involvement has sparked ethical concerns regarding the sharing and ownership of data. This study aims to develop a locally governed framework for ethical healthcare data exchange, establish an evidence base to understand local challenges in data transfer, and to identify potential solutions for data sharing with international research teams.

### Method

This cross-sectional qualitative study was conducted in Kathmandu, Nepal, using 11 multiple-choice and 12 open-ended questionnaire models. We conducted a pre-structured questionnaire survey to best identify local ethics issues related to international data transfer and proposed solutions for these challenges. The key representatives identified from the non-governmental and not-for-profit research institute ($n = 14$) and the life sciences society ($n = 7$) were invited to one-to-one blind interviews, and their recorded transcripts were coded using the QDA Miner Lite software (version 3.0) for analysis.

### Result

The ratio of female to male participants was 2:3, while the ratio of junior-level staff to senior staff (≥3 years of experience in the sector) was 1:9. Approximately 42.86% of

**Data availability statement:** All relevant data are within the paper and its Supporting information files.

**Funding:** Dr. Anurag Adhikari recieved the sub-grant award from Public Health Alliance for Genomic Epidemiology (PHA4GE) for the promote sustainable development in ethics and data sharing to support public health (REFERENCE: PHA4GE/2022-02). Additionally, there is no role of funder in study design, data collection and analysis, decision to publish, or preparation of the manuscript. (https://pha4ge.org/).

**Competing interests:** The All authors have declared that no competing interests exist.

**Abbreviations:** BSN, Biotechnology Society of Nepal; daTEF; develop the Data Transfer Ethics Framework; ERB, Ethical Review Board; INGO, International non-governmental organizations; IP, intellectual property; IPR, Intellectual Property Right; MTA, Materials transfer agreement; MoU, Memorandum of Understanding; NGO; non-governmental organizations; KIAS, Kathmandu Institute of Applied Sciences; KIIs, key informants; IP, Intellectual Property; IPR, Intellectual Property Rights; NBA, Nepal Biotechnology Association; PRI, Phutung Research Institute; RIBB, Research Institute for Bioscience and Biotechnology.

participants shared both raw and analytical data, while <5% shared no data with collaborators. Concerning knowledge, attitudes, and practices, most (38.46%) preferred open-access storage, while approximately 23.1% had limited knowledge, and 15.38% opted for confidentiality. Additionally, <10% were in the learning process and sought training in data transfer procedures. Within this group of key representatives, participants faced main challenges in the data transfer process from four key categories: (i) the lack of standardized guidelines from government or institutes for data transfer, (ii) inadequate awareness and training in data sharing, (iii) problems related to data sharing, and (iv) problems related to biological sample transfer.

## Conclusion

In summary, this study emphasizes the importance of a standardized data-sharing platform, focusing on protecting intellectual property rights and establishing a centralized data repository in Nepal. It also recommends educational reforms, legal measures, well-defined agreements, and dedicated oversight to ensure data integrity and security, while streamlining sample transfer processes to enhance transparency and scientific progress in Nepal's research landscape.

## Introduction

Data sharing in research is a fundamental practice that involves making research data accessible to other researchers within the same project or to the broader scientific community [1,2]. This practice is vital for several reasons, the most significant being its role in promoting transparency in research [3]. Transparency is a cornerstone of the scientific method, and data sharing is a tangible manifestation of this principle [3]. Data sharing allows researchers to access and scrutinize data, enhancing the credibility of research outcomes by verifying methods and results, while enabling multiple research groups to work together, often transcending geographical and disciplinary boundaries, resulting in more robust research and innovative solutions to complex scientific problems [4].

Despite the undeniable benefits of data sharing, there are challenges associated with transferring data within institutes and research groups, particularly in resource-scarce settings. Among many, data security, ethical considerations, and intellectual property rights are prominent issues [5,6]. To address these challenges, standardized guidelines and legal frameworks are crucial. These should encompass data security protocols, ethical considerations, and intellectual property agreements, providing legal protection and recourse in case of disputes or breaches [7–9]. Although international standards for data transfer and harmonizing data protection laws can simplify this process and reduce the risk of breaches or disputes, they are far from one-size-fits-all.

Historically, limited resources and restricted access to international scientific communities have hindered the research ecosystem in Nepal. However, in the past decades, Nepal has made significant progress in research and scientific inquiry [10],

along with the substantial growth in nationwide research institutions, and universities. Despite this promising evolution, limited funding, inadequate access to cutting-edge research equipment, and high-tech infrastructure constraints have impeded the progress of the Nepalese scientific inquiry process [11]. This inadequacy in infrastructure and funding in Nepal has led to unmet healthcare innovations, negatively impacting the quality of healthcare available to the population. The lack of government incentives for healthcare innovation, coupled with inadequate funding, has prompted non-Nepal-based international researchers to step in, providing funding and expertise for healthcare data and innovation [12]. While the increased presence of international researchers brings benefits, it also presents distinct challenges regarding data transfer and thus the ownership of research outcomes, specifically for non-profit and non-government research institutes that rely on international funding and grants to sustain. These issues include concerns about data security, ethical issues, and the rights tied to intellectual property with international funding bodies and/or collaborators [2]. Addressing these challenges may involve the establishment of standardized guidelines and legal frameworks, either at the institutional or governmental level [11]. However, there is no existing public data transfer guideline that outlines the specific challenges faced by researchers in non-governmental, non-profit organizations working in Nepal.

Given the scarcity of publicly available data on data transfer, this study aimed to develop a locally governed framework for ethical healthcare data exchange by employing a participatory-based approach using a structured questionnaire survey to establish an evidence base for understanding local challenges in data transfer and to identify potential solutions for data sharing with international research teams. The strength of this approach lies in engaging individuals with direct experience in various managerial and research roles within multinational research conducted in Nepal.

## Methodology

### Study design

This cross-sectional, qualitative study was conducted in non-governmental, non-profit research institutes, life sciences societies in Kathmandu, and among key stakeholders involved in Nepal's COVID-19 pandemic response. The study was conducted between 15th July and 30th October 2023. To develop the Data Transfer Ethics Framework (daTEF), this qualitative study was conducted in two stages: a scoping stage and a structured questionnaire survey stage (Fig 1).

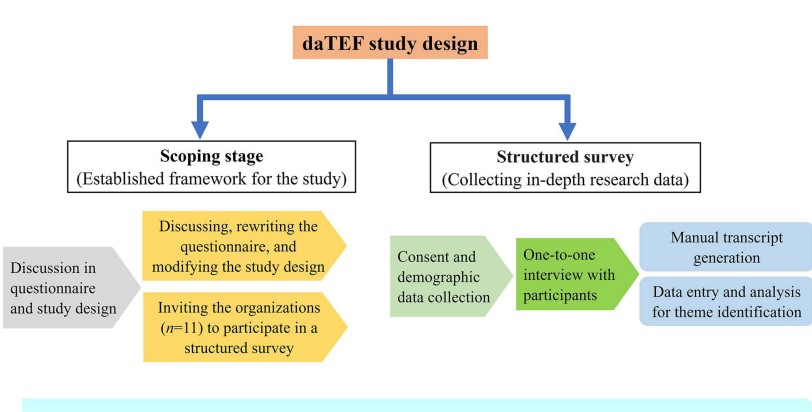

**Fig 1. Schematic representation of the daTEF study methodology.**

## A. Scoping stage

During the preliminary scoping stage, invitations were extended to the Executives/Directors of non-governmental non-profit research institutes ($n = 7$) and life sciences societies ($n = 4$) which were operating within Nepal. Once consented by the invitees (S2 and S3 Files), key experts with significant contributions in the field of biomedicine, comprising predominantly established researchers in Nepal, were selected at random from these non-governmental research institutes and the life sciences society, to attend a pre-study workshop. During this pre-study workshop, the investigator team and participants interacted virtually and/or face-to-face to assess the research questions and associated study critical designs [13]. After the workshop, participants offered suggestions for the questions and study design based on their critical evaluations.

## B. Structured survey stage

The survey stage utilized a pre-structured questionnaire research technique to gather and distill the insights and opinions of a panel of experts or stakeholders identified from the scoping stage. The interview was held blindly between the participants [13]. The method involved one round of blind interviews combined with questionnaires, incorporating controlled feedback to achieve a consensus or convergence of opinions against the identified pillar.

### Study participants

Based on the feedback of this scoping stage, we invited 11 organizations directors/presidents [(research institutes ($n = 7$) and life science societies ($n = 4$)] to nominate 5–10 key informants (KIIs) from each organization based on the criteria of having co-authored at least one scientific paper and/or engaged in a co-project with an international partner in last five years. Among them, 21 participants was enrolled in this study from different 5 organizations (Table 1). All consenting participants were invited to participate in a blind interview using a pre-structured questionnaire.

### Study tool

The study team developed a study questionnaire consisting of 11 questions to collect participants' socio-demographic data as a quantitative part. The quantitative part of the questionnaire was followed by the qualitative components, which included 12 open-ended questions (S4 File) that were asked during interviews with the KIIs.

### Data management and translation

Interviews were recorded through Zoom (version 5.10.1, 4420). Subsequently, each recording was transferred to the project computer in the form of an audio file. Following the deidentification process of these audio files, manual transcription was carried out. The original file, containing the audio recordings, was securely stored on the project computer, protected by encryption and a password.

Given that all interactive sessions and interviews were conducted in the Nepali language, all collected data underwent a manual translation process into English during the analysis phase. To ensure the quality of the transcripts, a three-step verification process was implemented. In this process, the initial transcripts for each KII were assessed independently by two members of the study team, without prior knowledge of each other's assessments. Subsequently, a third study team member performed a blind cross-validation, comparing both transcripts against the original audio files. The translated transcripts were coded using the QDA Miner Lite software (version 3.0) for the identification of similar thematic areas based on repeated words during analysis.

### Ethical approval

Before data collection, ethical approval was obtained from the Ethical Review Committee (ERB) of the Nepal Health Research Council (reference no. 3865, protocol registration no. 242/2023). The study was conducted in accordance with

the ethical principles and guidelines outlined in the Declaration of Helsinki. All participants were provided with a Participant Information Sheet (PIS) and gave written informed consent before participation. After reviewing the PIS and agreeing to take part as volunteers, participants were interviewed using a pre-structured questionnaire to record their responses.

## Results

### Demographics, work nature, and affiliation of the participants

Among all invited, twenty-one individuals affiliated with the research institute ($n = 14/21$, 66.67%) and life science societies ($n = 7/21$, 33.34%), that were operating in three different districts (Kathmandu, Bhaktapur, and Lalitpur) of Bagmati province of central Nepal consented for the study (**Table 1** and **Fig 2**).

The participant distribution reflected a female-to-male ratio of 2:3. With respect to professional seniority, the ratio of junior-level staff to senior staff (≥3 years of field experience) was 1:9. The largest professional group identified as biotechnologists (38%), followed by microbiologists (14%) and biochemists (14%). In contrast, disciplines such as public health, biomedical engineering, paramedicine, environmental science, and administrative management accounted for a smaller proportion of participants (**Fig 3**).

Most participants engage in collaborations with universities (76.1%), followed by research institutes (23.8%) and international funding agencies (23.8%) (**Fig 4A**). The primary drivers behind these collaborations include addressing the challenges of insufficient national funding, limited access to advanced research equipment, and a shortage of subject matter experts. Additionally, researchers also seek collaborations to acquire fresh insights, enhance their methodologies, gain a deeper understanding of region-specific issues, and fulfill the stipulations of funding agencies (**Fig 4B**).

### Participants experiences and practices related to data sharing

Regarding knowledge, attitudes, and practices toward data sharing, the largest proportion (38.46%) believed that data should be stored in an open-access format. Approximately one-quarter (23.1%) indicated limited knowledge of the data-sharing process, while 15.38% considered that research data should remain confidential. In addition, <10% of participants were still in the process of learning and expressed a need for formal training in research data transfer procedures (**Fig 5A**). Overall, 42.86% of participants reported sharing both raw and analytical data during the data-sharing process, whereas <5% did not share any data with their collaborators (**Fig 5B**).

### Research data transfer in Nepal: Insights from the four pillars of challenges

Among Nepalese researchers, we identified that the process of data transfer poses several intricate challenges, each potentially interconnected with others. Organized into four distinct pillars, our findings shed light on the obstacles researchers face, offering potential solutions for smoother data transfer processes. From the absence of standardized guidelines to issues of awareness, data sharing, and sample transfer, this exploration uncovers key insights crucial for

Table 1. Types and numbers of invited stakeholders and respondents in the structured questionnaire survey stage.

| Stakeholder group | Invited organizations to nominate respondents for the pre-structured questionnaire survey. (*n*, %) | Organizations that nominated respondents for the pre-structured questionnaire survey (*n*, %) | Participants who consented to the interview (*n*, %) |
|---|---|---|---|
| Non-governmental non-profit research institute | 7 (63.63%) | 3 (60%) | 14 (66.67%) |
| Life Sciences Society | 4 (36.36%) | 2 (40%) | 7 (33.34%) |
| **Total** | **11 (100%)** | **5 (100%)** | **21 (100%)** |

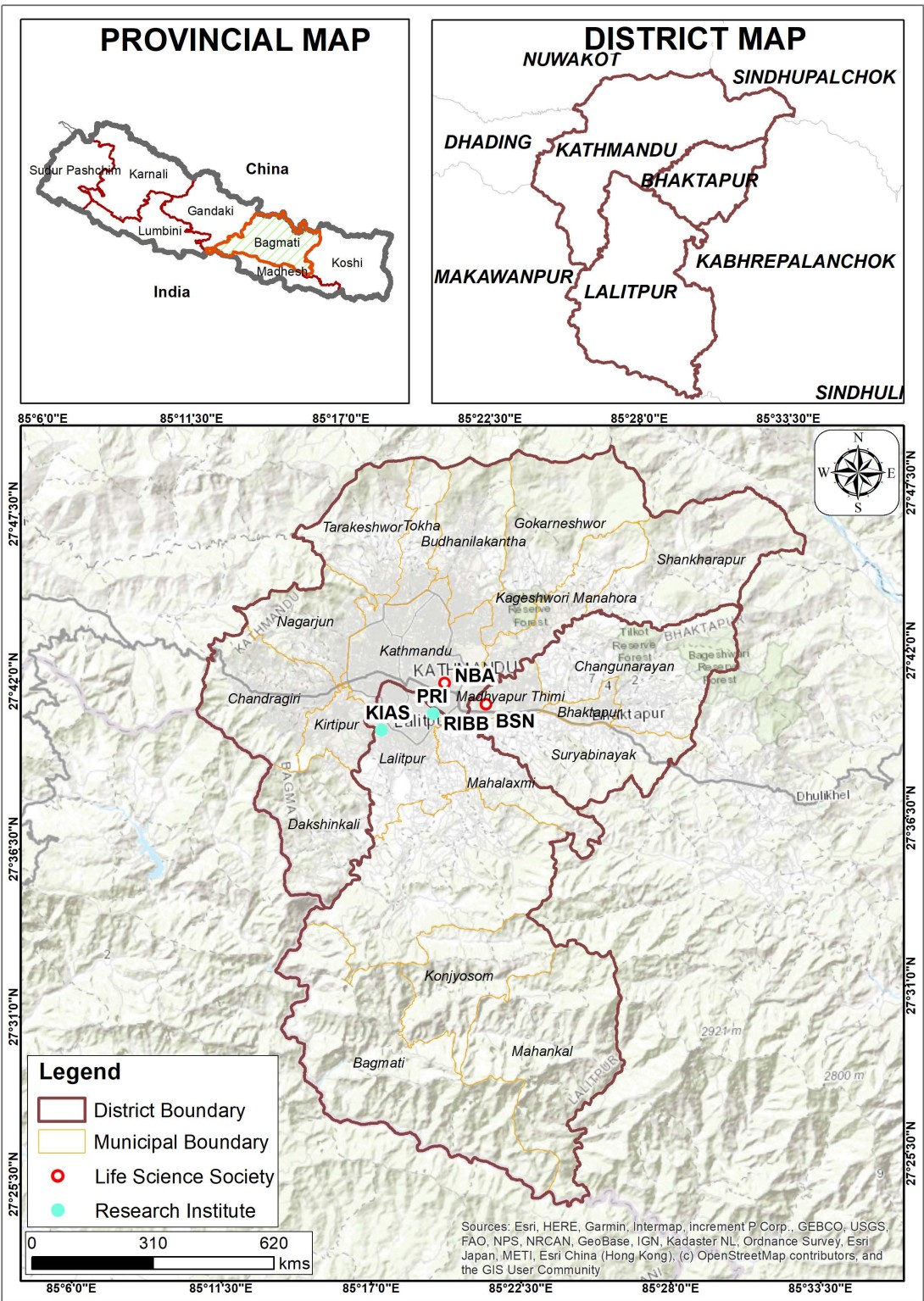

**Fig 2. Map showing the locations of research institutes (*n*=3/5) and life science societies (*n*=2/5) located in Kathmandu, Bhaktapur, and Lalitpur of Bagmati province.** NBA: Nepal Biotechnology Association; KIAS: Kathmandu Institute of Applied Sciences; PRI: Phutung Research Institute; RIBB: Research Institute for Bioscience and Biotechnology, BSN: Biotechnology Society of Nepal. The map was drawn by the authors. This map was created using ArcGIS (version 10.8.1) from Esri (www.esri.com), USGS, ©OpenStreetMap and contributors, Creative Commons Attribution 4.0 (CC BY 4.0).

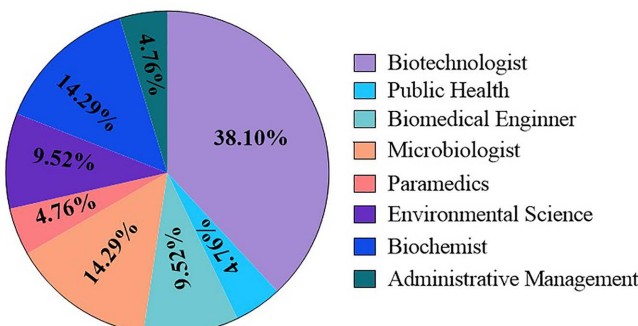

**Fig 3. Distribution of study participants (*n*=21) by professional background.**

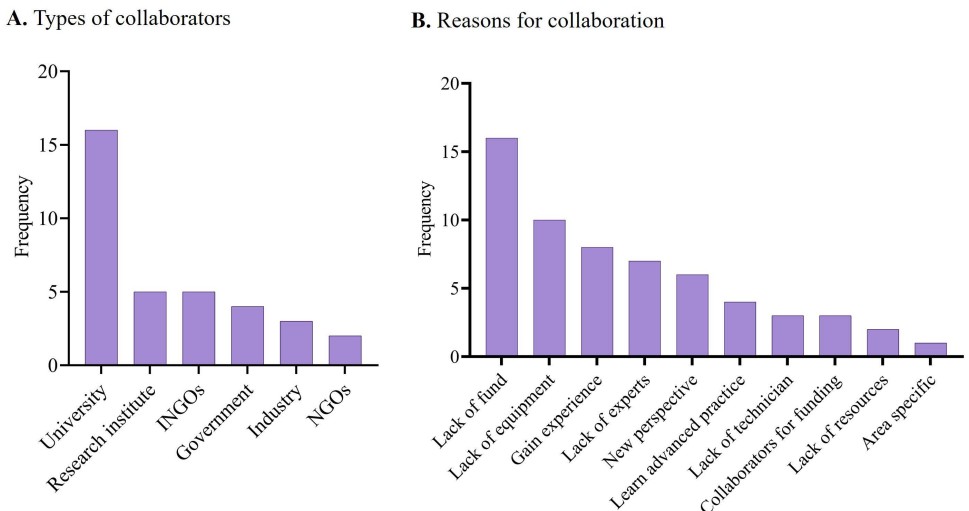

**Fig 4. Distribution frequencies of collaboration types (A) and reasons for collaborations (B).** INGO: International Non-Government Organization; NGO: Non-Government Organization.

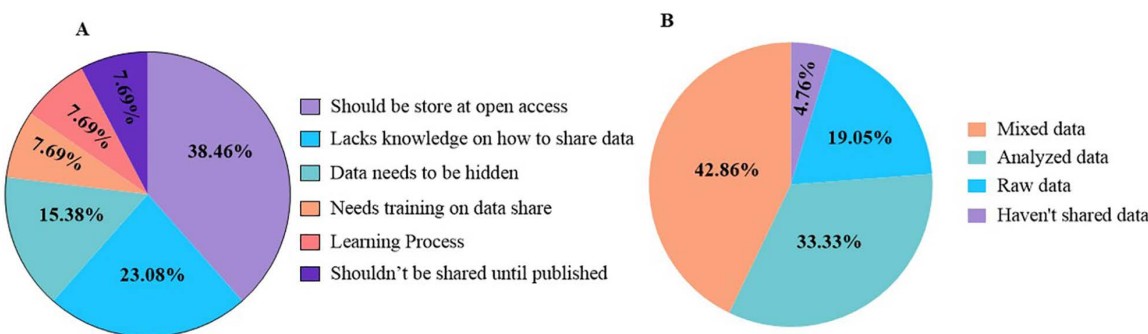

**Fig 5. Distribution rates of knowledge, attitude, and practices on data sharing (A) and types of data sharing (B).**

**Table 2. Key challenges experienced by scientific researchers in data sharing with international collaborative research teams.**

| Pillars of data sharing practices | Identified Challenges | Possible solutions and recommendations |
|---|---|---|
| I. The lack of standardized guidelines from the government or institutes for data transfer | 1. Lack of framework from the government | • The government should appoint scientific background employees to standardize the framework to prevent data that are national treasures.<br>• An ideal guideline without loopholes that can't be misused to send valuable data or samples that are in the nation's interest. |
| | 2. Lack of government departments and laws that protect intellectual property (IP) | • The government lacks proper and separate departments and laws to look over scientific research.<br>• New rules and regulations must be implemented and updated based on changing scenarios around the world.<br>• Nepali researchers should have equal rights to fight for their research and IP.<br>• A sense of security to researchers that their research and patents are protected. |
| | 3. Lack of a national depository | • Almost every nation has its own data depository to regulate and store the nation's data.<br>• The government should establish an open national data depository to store all of Nepal's research findings.<br>• All the nation's healthcare data and biological data should be stored and preserved for future research. |
| | 4. Lack of an independent department that oversees research | • The government should form an independent department that checks and regulates if the proposed approval and work are correspondent.<br>• A regulatory body to oversee what results and samples are being shared. |
| II. Inadequate awareness and training in data sharing | 1. Lacks knowledge of data sharing. | • The government should make standard laws to impose how and what data should be shared. |
| | 2. Requires special training involving data sharing | • The government should take responsibility for training new students and researchers. |
| | 3. Data shouldn't be shared | • The education system needs to be updated and teach about IP, patents, authorship, and ownership. |
| III. Problems related to data share | 1. Misused, manipulated, stolen. | • The government should implement strict rules to protect researchers' data. They need to enforce the law that protects Intellectual Property Right (IPR) and prevents others from stealing it.<br>• The organization and collaborator should have a clear goal and should clarify the percentage of patents and authorship. |
| | 2. Conflict between two parties | • Both parties should sign an agreement that clearly states the ownership and right to the research.<br>• An agreement should be signed at the initial phase of the contract to ensure that if the data is further used in the future, either permission is required, or recognition is given. |
| | 3. Uncertainty of data in the future, including plagiarism and stealing of data | • Strict laws should be implemented if plagiarism and the stealing of data are found.<br>• A separate governmental department should be established to check and protect data. |
| IV. Problems related to sample transfer | 1. Scientific samples from foreign countries aren't allowed in Nepal. | • The government should update and enforce material agreements to bring samples from nations with standard documentation.<br>• The government should allow the researcher to research resources that aren't available in our country |
| | 2. Challenges related to temperature maintenance during sample transfer, including limited options and the absence of proper facilities at customs duty. | • The government should expand the option for material transfer with more long-lasting temperature-stable methods.<br>• Develop a separate room that maintains the temperature to preserve the sample when transferring or receiving the sample. |
| | 3. Too much unnecessary paperwork delays sample transfer | • Make a standard, convenient protocol that can be carried out online. |
| | 4. Lack of a framework to transfer samples. | • The government should update the material transfer agreement and create a standard framework that is convenient for local researchers and foreign researchers to collaborate. |
| | 5. Limited courier option | • The government should be flexible to allow researchers to choose their desired courier for faster and more reliable means to transfer samples. |

advancing research in Nepal and other developing countries. The findings of key challenges and possible solutions of this participatory approach study are summarized in four distinct pillars, as presented in the following paragraph and **Table 2**.

**Pillar I: The lack of standardized guidelines from the government or institutes for data transfer.** It was observed that Nepal's government prioritizes research and research facilities. However, scientific researchers are significantly affected by the lack of a comprehensive framework provided by the government. The law and the protocols governing research and data transfer are outdated and not standard. This has made it difficult for researchers to obtain approval, ultimately causing delays in the research process. Furthermore, Nepal lacks a separate centralized governance department for research and intellectual property (IP) protection. The government needs to update and provide relief to the researchers by making better policies around IP protection laws. Additionally, Nepal also lacks data depository sites to store the data generated by Nepali researchers.

**Pillar II: Inadequate awareness and training in data share.** Study participants have shown concern and issues regarding Nepalese attitudes towards data sharing. While the majority of participants (38.46%) have displayed a positive attitude towards data sharing, some have encountered researchers who either lack knowledge of data sharing or proper training. With further interviews, the source of such behaviors lies in a lack of education and proper training. Even those who show enthusiasm are often unaware of which data needs to be protected and kept confidential. However, with the increasing trend of storing data as open access, attitudes towards data sharing are improving.

**Pillar III: Problems related to data sharing.** Participants who shared the data have either personally experienced or heard about the challenges faced by other researchers when sharing data. Commonly reported issues include misuse, manipulation, and plagiarism. The rise of these problems can be attributed to the absence of laws and IP departments in the country dedicated to protecting research and research data.

**Pillar IV: Problems related to biological sample transfer.** One of the major challenges encountered by Nepalese researchers is the transfer of samples. Some participants have described it as one of the most complex, time-consuming, expensive, and risky challenges. The demotivation stems from the lack of a proper, outdated framework on Material Transfer Agreement (MTA), protocol, temperature-maintaining methods, and limited courier options. Furthermore, the process of importing research samples from foreign countries needs to be updated with proper paperwork and accessible means to apply for it.

### Guidelines to aid in identifying and overcoming data-transfer challenges

In response to the identified data-transfer challenges, stakeholders were engaged in interviews to provide insights and recommendations for addressing four major issues. This collaborative effort culminated in the development of a succinct guideline (S1 File). This guideline presents a framework for understanding the context for data transfer within the non-government and non-profit research ecosystem of Nepal, considering the unique challenges, opportunities, and associated solutions that exist in this resource-scarce setting.

The guideline begins by acknowledging the obstacles posed by the absence of standardized protocols and government support, recognizing that these factors can hinder the sharing of research data. As a solution, it strongly advocates for the establishment of a national framework, an initiative that is best suited for government intervention. This guideline helps identify that safeguarding intellectual property rights by establishing a centralized data repository within institutes/government bodies could be pivotal for data protection and accessibility in the future. The guideline further emphasizes the necessity of reforming the educational landscape to encompass essential aspects like IP, patents, authorship, and ownership. It strongly recommends that the government play a proactive role in organizing training programs to equip future researchers with the skills and knowledge required for ethical and efficient data sharing.

The guideline extends its coverage to address issues intrinsic to research data sharing, including concerns related to data misuse and conflicts among parties involved. To address these problems, it advocates for the implementation of rigorous legal measures that protect the integrity of researchers' data. In addition, it emphasizes the importance of

well-defined agreements of material transfer agreement (MTA) and Memorandum of Understanding (MoU) between collaborating parties to delineate data ownership and rights over shared research data before the start of the study itself. Furthermore, the guideline proposes the creation of a dedicated government department responsible for overseeing and securing research data, thus enhancing data integrity and overall security.

The guideline also delves into the complexities surrounding sample transfer, particularly the restrictions on importing scientific samples from foreign countries into Nepal. As a solution, it suggests the revision and enforcement of material transfer agreements, facilitating cross-disciplinary international collaboration while adhering to standardized documentation. Furthermore, the guideline highlights the challenges associated with maintaining sample temperatures during transfer and recommends the establishment of temperature-controlled facilities and standardized protocols to address these issues. Cumbersome paperwork-related delays in research sample transfer should be addressed through the implementation of standardized, user-friendly, and online protocols, alongside the need to diversify courier options for researchers, affording them the flexibility to choose the most dependable means of sample transfer.

By adopting these recommendations, researchers based in Nepal can establish an efficient and secure data transfer process, fostering greater transparency, collaboration, and scientific progress within its international research landscape.

## Discussion

Data sharing is a critical practice in research, as it promotes transparency, collaboration, and innovation in the scientific community [14]. The transparency it provides allows for the verification of research methods and results, reducing the likelihood of erroneous or fraudulent findings [15]. Furthermore, collaborative research across geographical and disciplinary boundaries is facilitated, leading to more robust research and innovative solutions. While the benefits of data sharing are undeniable, the process of transferring data on a global scale presents various challenges, which require standardized guidelines and legal frameworks to address effectively [9,16].

Despite limited knowledge of data transfer, most respondents in this study reported engaging in data transfer with their collaborators or funding agencies. The study further highlights the unique challenges faced by Nepalese researchers in this context. The study findings indicate a need for addressing data transfer challenges, especially within the Nepalese research context. The lack of standardized guidelines and government support for data transfer poses a significant hindrance for Nepalese researchers, and this issue is not local but very relevant globally [17]. Outdated laws and protocols affect the approval process and delay Nepalese research projects, as this was reported by numerous scholars in the past [6,16,18,19]. These calls are for the Nepalese government and/or research institutions to update policies related to intellectual property protection and establish a centralized data repository. Internationally, the adoption of a centralized regulatory body for funding and oversight has been prevalent [20–23], indicating a recurring need among Nepalese researchers. Additionally, educational reforms and training programs are necessary to equip future researchers with the skills and knowledge required for ethical and efficient data sharing [24].

The study also highlights issues related to data sharing, including misuse, manipulation, and plagiarism, arising from the absence of laws and intellectual property departments to protect research data [25,26]. Robust legal measures are recommended to protect the integrity of researchers' data, along with well-defined agreements between collaborating parties to clarify ownership and rights over shared research and findings.

In Nepal, sample transfer is a particularly complex, time-consuming, expensive, and risky process [27]. The lack of proper frameworks, such as Material Transfer Agreements (MTAs) and temperature maintenance methods, hinders efficient sample transfer [28]. Additionally, importing samples from foreign countries requires streamlining the paperwork and providing an accessible means to apply for it. This study's recommendations include revising and enforcing material transfer agreements, establishing temperature-controlled facilities, and implementing standardized, user-friendly, and online protocols to streamline the sample transfer process.

The development of a comprehensive guideline to address these challenges is a significant step toward enhancing data transfer processes in Nepal. It emphasizes the importance of government intervention in creating a national framework to safeguard intellectual property rights, establish a centralized data repository, and provide essential education and training for researchers. Furthermore, it highlights the need for legal measures to protect data integrity, well-defined agreements, and dedicated government departments to oversee and secure research data. Diversifying courier options for sample transfer is also recommended to offer researchers greater flexibility.

## Conclusion

This study identifies gaps in ethical awareness and the absence of standardized data-sharing protocols. The proposed guidelines offer a roadmap for Nepal to overcome the challenges in research data sharing and transfer for ethical and sustainable research ecosystem. This study also contributes a new dimension to the global discourse on ethical data sharing governance that bridges the gap between global research and local realities by promoting mutual accountability and equitable partnerships, and shared responsibility in international scientific collaboration. By addressing these issues, Nepal can foster greater transparency, international collaboration, and scientific progress within its evolving research landscape, ultimately contributing to advancements in various fields, including healthcare. This not only benefits the research community but also has the potential to improve the quality of healthcare services and innovation in the country. The lessons learned in Nepal can also serve as a valuable reference and critical evidence for other regions facing similar challenges in data management and sharing for research collaboration.

## Limitations of the study

This study was limited to non-profit life sciences research institutes located in Kathmandu, the capital city of Nepal. The findings underscore the challenges faced by researchers in Nepal's non-profit research institutes. To our knowledge, it is the first to systematically examine data transfer practices and challenges in Nepal's research context, engaging participants with direct experience in multinational collaborations. By employing a participatory approach, the study provides valuable insights into local practices, knowledge gaps, and barriers, which can inform the development of more ethically and locally grounded data-sharing frameworks in resource-limited settings. However, these outcomes cannot be generalized to all low- and middle-income countries (LMICs). Further studies with larger sample sizes are needed to strengthen the evidence supporting the guidelines developed in this study.

## Supporting information

**S1 File. Guidelines for data transfer in research.**
(PDF)

**S2 File. Consent form-daTEF study.**
(PDF)

**S3 File. Participant information sheet daTEF study.**
(PDF)

**S4 File. Questionnaires form (daTEF).**
(PDF)

## Acknowledgments

We gratefully acknowledge the research participants for their time, contributions, and insights from various research institutions. Consent was sought from each participant to publish the results.

## Author contributions

**Conceptualization:** Gopiram Syangtan, Minu Singh, Binod Rayamajhee, Anurag Adhikari.

**Data curation:** Gopiram Syangtan, Sauhardra Manandhar.

**Formal analysis:** Gopiram Syangtan, Sauhardra Manandhar, Manjula Bhattarai, Binu Shah, Binod Rayamajhee.

**Funding acquisition:** Anurag Adhikari.

**Investigation:** Gopiram Syangtan, Sauhardra Manandhar, Minu Singh.

**Methodology:** Gopiram Syangtan, Sauhardra Manandhar, Minu Singh, Anurag Adhikari.

**Project administration:** Gopiram Syangtan.

**Resources:** Anurag Adhikari.

**Supervision:** Minu Singh, Binod Rayamajhee, Anurag Adhikari.

**Writing – original draft:** Gopiram Syangtan, Manjula Bhattarai, Binu Shah, Binod Rayamajhee, Anurag Adhikari.

**Writing – review & editing:** Gopiram Syangtan, Manjula Bhattarai, Binu Shah, Binod Rayamajhee, Anurag Adhikari.

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
