## [Decision Letter · Decision Letter 0]

24 Jun 2025

Dear Dr. Rayamajhee,

Please revise the manuscript based on peer reviewer's comments and mention NHRC ethical approval reference number in the ethical approval section. I found you have mentioned the information in the online submission. I request you to mention that in the paper too. 

We look forward to receiving your revised manuscript.

Kind regards,

Kanchan Thapa, MPH, MPhil

Academic Editor

PLOS ONE

4. We note that Figure 2 in your submission contain [map/satellite] images which may be copyrighted. All PLOS content is published under the Creative Commons Attribution License (CC BY 4.0), which means that the manuscript, images, and Supporting Information files will be freely available online, and any third party is permitted to access, download, copy, distribute, and use these materials in any way, even commercially, with proper attribution. For these reasons, we cannot publish previously copyrighted maps or satellite images created using proprietary data, such as Google software (Google Maps, Street View, and Earth). For more information, see our copyright guidelines: http://journals.plos.org/plosone/s/licenses-and-copyright.

Reviewers' comments:

Reviewer's Responses to Questions

**Comments to the Author**

1. Is the manuscript technically sound, and do the data support the conclusions?

Reviewer #1: Yes

Reviewer #2: Yes

Reviewer #3: Partly

2. Has the statistical analysis been performed appropriately and rigorously?

Reviewer #1: I Don't Know

Reviewer #2: Yes

Reviewer #3: Yes

3. Have the authors made all data underlying the findings in their manuscript fully available?

Reviewer #1: Yes

Reviewer #2: Yes

Reviewer #3: No

4. Is the manuscript presented in an intelligible fashion and written in standard English?

Reviewer #1: Yes

Reviewer #2: Yes

Reviewer #3: Yes

Reviewer #1: This manuscript presents a study on ethical data transfer practices, with particular relevance for low- and middle-income countries navigating research collaborations. The development of a Data Transfer Ethics Framework is a notable contribution, and the study is clearly written, methodologically appropriate, and well-structured. It could benefit from minor revisions to improve clarity, address inconsistencies, and strengthen the link between data and conclusions.

My main comment is that I thought Delphi methodology involved repeated interviews to reach consensus and that this process, if adopted, is not clearly articulated in the manuscript.

The link between findings and the final framework could be more explicit

Reviewer #2: The paper is well written, considering critical aspects in research including gaps identified in the literature review to understand potential issues in data transfer and the possible solutions. The study obtained ethics approval from relevant Board. The result synthesis into key thematic areas also makes it easier for readers to understand key findings, as well as the table summary of challenges experienced by scientific researchers in data sharing and the possible solutions. The study could improve by addressing some of the possible study limitations as well as providing generalization limit. For example, whether the study is only applicable to Nepal or can be generalized to a larger population (which should be specified).

Reviewer #3: Dear Editor,

Thank you for the opportunity to review the manuscript, which outlines the development of a data transfer ethics framework through the elicitation of experts’ opinions. The authors have identified four principal challenges in the data transfer process and provided potential solutions. While the idea and the findings hold considerable value for the Nepalese research community, a significant concern regarding methodological clarity warrants attention:

The authors outlined two stages of the study based on the methods and Figure 1:

1. The initial "Scoping" stage involved inviting executives and directors from 11 organisations to participate in a workshop with the investigator team to assess the research question and study design and to provide suggestions based on their expert evaluation.

2. The second "Delphi" stage involved experts nominated by the panel from the first stage, who underwent a single round of anonymous interviews, questionnaires, and controlled feedback against the identified pillars.

The authors labelled the study as a modified Delphi technique; however, they reported undertaking only one round of blinded interviews and questionnaires. The fundamental structure of the Delphi technique and its variants involves two or more rounds of questionnaires (or the option to respond at least twice) with controlled feedback presented starting at the second round, according to the guidelines. The authors reported in lines 107-108 that: "The Delphi stage utilized a structured and iterative research technique, etc." The term "iterative" indicates multiple rounds, which contradicts the description of a single round in lines 109-111: "The methods involved one round of blinded interviews, etc."

The statements are contradictory; if the study involved a single round of expert opinions, the characteristics of the study more resemble an expert opinion survey or a structured survey rather than a Delphi survey due to the lack of the iterative nature of the Delphi technique. The methods section needs clarification and subsequent revision of the manuscript.

Additionally, the authors did not highlight any limitations to the study and whether these limitations were based on the study participants (experts gathered), the actual study methods, or generalisability of the findings.

Kind regards,

**Do you want your identity to be public for this peer review?** For information about this choice, including consent withdrawal, please see our Privacy Policy

Reviewer #1: No

Reviewer #2: **Yes: ** Fahad Alenezi

Reviewer #3: No

---

## [Author Response · Author response to Decision Letter 1]

1 Aug 2025

Dear Editor,

We sincerely thank you and the independent reviewers for taking the time to review our manuscript and for providing valuable comments and suggestions that have helped improve the quality of our work. We appreciate the opportunity to revise and strengthen the manuscript in response to the reviewers’ feedback.

All comments have been carefully addressed, with corresponding point-by-point responses provided for each. We have submitted both the tracked-changes version and the clean version of the revised manuscript, along with the author response letter. Thank you.

Best regards,

Dr Binod Rayamajhee, on behalf of all authors

UNSW, Sydney, Australia

b.rayamajhee@unsw.edu.au, binod@kribs.org.np

---

## [Editor Report · Decision Letter 1]

24 Aug 2025

Dear Dr. Rayamajhee,

Thank you for submitting your manuscript to PLOS ONE. After careful consideration, we feel that it has merit but does not fully meet PLOS ONE’s publication criteria as it currently stands. Therefore, we invite you to submit a revised version of the manuscript that addresses the points raised during the review process.

We look forward to receiving your revised manuscript.

Kind regards,

Kanchan Thapa, MPH, MPhil

Academic Editor

PLOS ONE

Journal Requirements:

Additional Editor Comments:

Ethical consideration

I suggest to rewrite the ethical consideration section. You added a line about Helsinki declaration, I suggest to rewrite the ethical consideration in original language what you exactly did.

I suggest to modify your Abstract and background section adding - What is the global significance of this study? Why people outside of country need to read your paper and what they can replicate or modify in their context?

Please proof read your paper and resubmit.

Results

Table 1. is your table complete? Please review other paper and presenting your table heading an appropriate way.

Figure2. Is this your table heading?

Figure 3:???

Figure 4:

I suggest to present socio-demographic findings in one table unless and until they really impact on the overall study findings.

Line 174: Despite limited knowledge of data transfer, most respondents were involved in data transfer?

Please write an appropriate section sub-headings. I think such information as of line 174 should go in the discussions section.

Thank you for adding the limitation section, but I think presenting your strength is equally important.

---

## [Author Response · Author response to Decision Letter 2]

28 Aug 2025

The Editor-in-Chief and Academic Editor, we would like to thank you for dedicating your valuable time to reviewing and editing this manuscript, offering comments and suggestions to enhance the quality of our work. We greatly appreciate your valuable comments and suggestions. We have submitted both the track change and clean version of the revised manuscript along with point-by-point responses to editor comments.

---

## [Decision Letter · Decision Letter 2]

2 Oct 2025

Dear Dr. Rayamajhee,

Thank you for submitting your manuscript to PLOS ONE. After careful consideration, we feel that it has merit but does not fully meet PLOS ONE’s publication criteria as it currently stands. Therefore, we invite you to submit a revised version of the manuscript that addresses the points raised during the review process.

We look forward to receiving your revised manuscript.

Kind regards,

Kanchan Thapa, MPH, MPhil

Academic Editor

PLOS ONE

Journal Requirements:

Reviewers' comments:

Reviewer's Responses to Questions

**Comments to the Author**

Reviewer #4: (No Response)

Reviewer #5: (No Response)

2. Is the manuscript technically sound, and do the data support the conclusions?

Reviewer #4: Yes

Reviewer #5: Partly

3. Has the statistical analysis been performed appropriately and rigorously?

Reviewer #4: Yes

Reviewer #5: Yes

4. Have the authors made all data underlying the findings in their manuscript fully available?

Reviewer #4: Yes

Reviewer #5: Yes

5. Is the manuscript presented in an intelligible fashion and written in standard English?

Reviewer #4: Yes

Reviewer #5: Yes

Reviewer #4: Thank you for submitting your manuscript.

Abstract

1. For methods, dates of the interview can be left out since they appear in the main Methods section.

2. For Results, can you summarize these cutting out explanations as you have given.

3. The Abstract should be summarized further, it is very long.

Introduction

What is the rate of unmet contraceptive use in Africa, East Africa (if these are documented) before you scale it down to Rwanda? A comparison is important for readers to know how critical the situation.

Methods

1. What sampling method did you use?

2. Can we have more about the inclusion criteria for the study participants? For example, how long should they have stayed in the study area? Or did you take on even those who had recently relocated to your study area?

3. No pretest of data collection instrument has been reported.

4. How did you take care of anonymity and confidentiality? For example, were the interviews in enclosed or open but private spaces? Where these chosen by the participants or the interviewer?

Results

Good.

Discussion

Good.

Conclusion

This seems so thin. Of what importance are the study findings regionally and internationally, not only in Nepal? Let this clearly stand out because it is the gist of what your study is adding to the field of research.

Reviewer #5: Thank you for the opportunity to review this manuscript. It addresses an important topic and offers useful insights. To strengthen the paper, I provide comments below aimed at improving clarity, consistency, and impact.

Abstract

1) (L22, L93): The manuscript describes the study design inconsistently, referring to it as both a cross-sectional qualitative study (L 22) and a mixed-methods approach (L 93). Please clearly define the study design and ensure consistency across the abstract and methods.

2) (L18-20, L 86-88): Aims differ between the abstract and introduction sections; one aims to develop a framework for ethical healthcare data transfer, while the other seeks to present recommendations based on reported local challenges. Please revise the objectives to reflect the actual contribution.

3) The abstract claims the framework is “essential for advancing global disease surveillance, strengthening outbreak response, optimizing patient care, and safeguarding privacy.” Yet, the introduction and discussion did not explain how these global impacts would be achieved. Please either expand on these links or adjust the claim to match the study’s scope.

Introduction

4) Line 75-78: Please provide references or specify details to substantiate these claims.

Results

5) Figure 3, 5 (A, B): Current figures are difficult to interpret without percentages. Please provide percentages in the charts or present the data in tables.

6) L175–176: Figure mislabeled, text refers to 5B but marked as 5A.

7) L180–181: Statement does not match the type of data sharing shown in Figure 5B.

8) L221: The heading should specify biological sample transfer.

9) L248–250: The text refers to the need for “well-defined agreements,” but it does not explain what constitutes such an agreement or what criteria should be included. Please clarify.

**Do you want your identity to be public for this peer review?** For information about this choice, including consent withdrawal, please see our Privacy Policy

Reviewer #4: No

Reviewer #5: No

---

## [Author Response · Author response to Decision Letter 3]

20 Oct 2025

We would like to thank you for dedicating your valuable time to reviewing and editing this manuscript, offering comments and suggestions to enhance the quality of our work. We greatly appreciate your valuable comments and suggestions. All comments are reproduced in the table (attached in the submission), with corresponding point-by-point responses provided for each.

Sincerely yours,

On behalf of all authors,

Binod Rayamajhee, MSc, PhD

---

## [Editor Report · Decision Letter 3]

26 Oct 2025

Development of Data Transfer Ethics Framework (daTEF): A participatory approach to delivering evidence-based guidelines for healthcare data transfer

PONE-D-25-14540R3

Dear Dr. Rayamajhee,

We’re pleased to inform you that your manuscript has been judged scientifically suitable for publication and will be formally accepted for publication once it meets all outstanding technical requirements.

Kind regards,

Kanchan Thapa, MPH, MPhil

Academic Editor

PLOS ONE
---

## [Editor Report · Acceptance letter]

PONE-D-25-14540R3

PLOS ONE

Dear Dr. Rayamajhee,

I'm pleased to inform you that your manuscript has been deemed suitable for publication in PLOS ONE. Congratulations! Your manuscript is now being handed over to our production team.

Kind regards,

on behalf of

Mr. Kanchan Thapa

Academic Editor

PLOS ONE